# Comparing Two Methods of Acute: Chronic Workload Calculations in Girls’ Youth Volleyball

**DOI:** 10.3390/sports11030051

**Published:** 2023-02-23

**Authors:** Claire Schumann, Monica Wojciechowski, Jennifer A. Bunn

**Affiliations:** 1College of Osteopathic Medicine, Sam Houston State University, Conroe, TX 77304, USA; 2College of Health Sciences, Sam Houston State University, Huntsville, TX 77341, USA

**Keywords:** athlete monitoring, youth athletes, team sports

## Abstract

Monitoring training load using acute:chronic workload ratio (ACWR) enables coaches to maximize fitness potential while mitigating injury risks by maintaining an optimal ACWR range. There are two methods of determining ACWR: rolling average (RA) and exponentially weighted moving average (EWMA). This study aimed to (1) compare weekly changes in kinetic energy (KE) output in female youth athletes (*n* = 24) during the high school (HSVB) and club volleyball (CVB) seasons and (2) evaluate the agreement in RA and EWMA ACWR calculations during the HSVB and CVB seasons. Weekly load was measured using a wearable device, and RA and EWMA ACWRs were calculated using KE. The HSVB data showed spikes in ACWR at the onset of the season and during one week mid-season (*p* = 0.001–0.015), but most weeks were in the optimal ACWR range. The CVB data had greater weekly variations throughout the season (*p* < 0.05), and many weeks were outside of the optimal ACWR range. There were moderate correlations between the two ACWR methods (HSVB: r = 0.756, *p* < 0.001; CVB: r = 0.646, *p* < 0.001). Both methods can be used as a monitoring tool for consistent training like that in HSVB, but more research is needed to investigate appropriate methods for an inconsistent season like that of CVB.

## 1. Introduction

Monitoring athlete conditioning and maintaining a balance between fitness and fatigue are at the forefront of sports research and training. One of the most common methods of managing athlete workload is using the acute:chronic workload ratio (ACWR). The purpose of this tool is intended to reduce the risk of injuries while maximizing fitness and balancing recovery [1]. ACWR is calculated as the acute workload divided by the chronic workload. Acute workload is typically measured daily over a seven-day period and represents fatigue in an athlete. Chronic workload is generally the average of 28 days of the workload and represents the fitness of an athlete [1,2]. The optimal ACWR is a range between 0.8 and 1.5—an ACWR above 1.5 increases the risk of injury, and a value below 0.8 may result in loss of fitness [1].

ACWR can be calculated using the rolling average (RA) model or the exponentially weighted moving average (EWMA) model. The RA model weighs chronic and acute workloads equivalently and does not account for variations in training schedules [1]. This method allows for a simple ratio calculation because the RA model does not consider the compounding effect of recent training volume. The EWMA model has been shown to be a better indicator of performance and injury because this model places greater emphasis on the most recent work an athlete performed and accounts for the decline in athlete fitness [3,4]. The EWMA calculation weighs the most recent weeks’ chronic workload more than that of the workload performed three weeks prior. This weighting allows for a reduced mathematical impact of the declined physiological work from three weeks prior.

Currently, most ACWR studies have been derived from professional, adult male athletes [1,2,3,4,5,6,7]. While ACWR has proven helpful in male sports, the research on female adolescent athletes is limited [5]. Studies have shown that using ACWR is effective in managing injury and training load in non-elite intercollegiate female soccer and rugby players (18–24 years) and in preventing load spikes that may result in injury in professional elite female basketball players (mean age: 21 years) [8,9]. Previous literature on elite female volleyball athletes showed that the ACWR EWMA stayed in the optimal performance zone for the duration of a competitive season [10]. These data are useful as foundational knowledge that may be applied to other volleyball populations but provide no insight into the training load endured by the athletes over an entire training year. Many studies investigating female sports rely on other monitoring methods (e.g., heart rate variability, ratings of perceived exertion) or suggest altering training methods (e.g., incorporating prevention exercises) over monitoring objectively measured training load [11,12,13,14]. While there is nothing inherently wrong with any of these methods, it is staggering that a well-known monitoring tool such as ACWR, in combination with objective measures, is rarely applied to female sports, much less adolescent, developing female athletes.

Volleyball is a sport that has been steadily growing in participation in the United States. In 2019, there were nearly half a million female adolescents participating in volleyball, with many of these athletes participating in both the high school and club volleyball seasons [15]. Athletes are expected to miss school as needed to attend tournaments, creating a stressful student–athlete relationship. Many high school–aged (14–18 years old) volleyball players play approximately 48 weeks a year between the high school season, club season, and summer training camps. In Texas (the state with the highest number of high school volleyball athletes), the high school season is about four months long, and athletes train or play games about six days a week [16]. During the club season, athletes typically train two days a week for almost eight months. Three of those months consist of heavy tournament play where athletes play 9–12 matches over a three-day period for several weeks. The remaining five months consist of light tournament play where athletes play six to eight matches over two days during each tournament, but tournaments are less frequent. Club tournaments often require traveling to a different city or state to compete against teams from all over the United States. As these athletes play volleyball almost year-round, measuring training volume will likely be very beneficial to ensure that they are not being overtrained and at risk for injury. Professional female volleyball players have an overall injury rate of 2.6 per 1000 playing hours, with ankle sprains cited as the most common injury [17]. In addition, volleyball is the eighth most injury-prone sport for young adults ages 14–20 years [18]. As most youth volleyball players are involved in both club and high school, overuse injuries are a great concern and exemplify the need for an accurate model of monitoring workload.

In Texas, high school girls’ volleyball is a year-round sport because of the demands from the club and high school [16] and incorporating ACWR as a tool for managing training and game volume may be beneficial to coaches and athletes. The primary aim of this study was to compare changes in external load, measured via kinetic energy, in female youth athletes during the high school and club volleyball seasons. Weekly load is evaluated using both the RA- and EWMA-ACWR calculations. The secondary aim is to evaluate the agreement in RA- and EWMA-ACWR calculation methods during the high school and club seasons. We hypothesized that ACWR would remain in the optimal zone for the high school season but would follow a more volatile pattern during the club season. We also hypothesized that RA and EWMA ratios would have a higher correlation during the high school season than the club season. To our knowledge, this is the first study to provide insight related to longitudinal training and game loads in high school volleyball athletes. This is also the first study to apply the concept of ACWR to this population. Further, this is the first study to provide objective data for a consistent training and game schedule (high school) and inconsistent schedule (club) within the same sport. The schedule differences are expected to produce variations in ACWR between the two types of seasons.

## 2. Materials and Methods

### 2.1. Study Design

This study utilized two different datasets that were prospective observational studies. The high school volleyball (HSVB) dataset was conducted with a 6A high school volleyball team in the 2021 University Intercollegiate League in Texas. The club volleyball (CVB) dataset was conducted with a Texas club volleyball team competing in USA volleyball tournaments during the bulk of the 2021–2022 season—the first 20 weeks of the 29-week season. Athletes participated in either the HSVB portion or the CVB portion, not both. This study was approved by the Sam Houston State University institutional review board (IRB-2021-202) and was conducted in alignment with the Declaration of Helsinki. All participants provided their assent, and their parents or legal guardians provided consent for study participation.

### 2.2. Participants

#### 2.2.1. HSVB Data

The included athletes were members of the high school varsity volleyball team participating in the team camp in July 2021 and had parent/guardian approval. An athlete was excluded from the study if the coach did not indicate them as a top-12 player on the team because of the limited number of devices used for measurement. A total of 12 athletes consented to this portion of the study. Athletes participated from July 2021 through October 2021. The season was divided into four phases: pre-season, tournament play, conference play, and playoffs. The two-week long pre-season (weeks 1–2) included a total of 9 practices and 12 sets played. Tournament play lasted five weeks (weeks 3–7) and included 13 practices and 68 played sets. Conference play spanned seven weeks (weeks 8–14), with 27 practices and 43 sets played. The playoff portion of the season lasted two weeks (6 practices and 18 sets played) but was not included in the dataset because of the inconsistent data collection. In total, the team participated in 40 matches and finished the season in the top eight in their division of 244 teams.

#### 2.2.2. CVB Data

The included athletes were members of the top team of the age group within the club, began the season in November 2021, and had parent/guardian approval for the study. One athlete was excluded from the study, owing to a late-season start related to an injury. A total of 12 athletes participated in this portion of the study. Athletes participated from November 2021 through April 2022. The season was divided into four phases: pre-season, prep tournaments, qualifier phase, and nationals. Pre-season was approximately six weeks long (weeks 1–6), with two practices per week and two one-day tournaments held locally on the weekend. The prep phase was also six weeks (weeks 7–12), with two practice per week, two two-day tournaments, and one three-day tournament. The qualifier phase was eight weeks long (weeks 13–20) and included two practices per week, one two-day tournament, and three three-day tournaments. The three-day tournaments in this phase were all USAV events where the team aimed to achieve a qualifying spot for nationals in the summer. The last nine weeks of the season—the nationals phase—were not included because of a less predictable schedule and the team competed in two tournaments.

### 2.3. Measures

Data were collected in both datasets using VERT model KMT devices (VERT Team System, Fort Lauderdale, FL, USA). These devices had been verified by third-party analysis for the accuracy of jump count and vertical jump [19]. Each athlete wore the device during all practices and games in which they participated. The devices were worn and used according to best practice standards developed by the device company. All data were processed by the VERT Team System App on an iOS device and uploaded to the myVERT cloud.

The data collected by VERT included session totals for the number of jumps, jumps completed greater than 50 cm, maximum jump height, movements per minute, stress percentage upon landing, and kinetic energy (J/lb). Kinetic energy was calculated within VERT according to the methods provided by Charlton et al. and included the final velocity of an athlete after an acceleration and the athlete’s mass [20]. This value represented the external workload of an athlete for a given session and was the primary variable used for analysis in the present study. Kinetic energy was selected as the primary variable because these data were longitudinal and included a total of 34 weeks observed, and it provided a concept of the overall training volume based upon movements and not jumps alone. It served as an optimal variable for evaluating both jumping and non-jumping positions across the game. This variable has been positively compared to the PlayerLoad amalgam variable used by Catapult Sports [21].

The ACWR for kinetic energy was calculated weekly in the HSVB and CVB datasets using the RA and EWMA. Both methods utilized a 7-day period for the acute workload and a 28-day period for the chronic workload. The calculation of the acute workload was the average workload during the most recent 7-day period, and ACWR values were calculated for the end of each week. The chronic workload using the RA method averaged the preceding 28-day workloads [1,2,4,7]. The chronic workload using the EWMA method utilized time decay to account for the most recent workload when calculating the chronic workload [4,22]. A time decay constant was determined using a 7-day workload and a 28-day workload to account for the acute and chronic workloads. The workload of the current day was multiplied by the degree of decay and then added to a value that enhanced the most recent training load. This method allowed time to be considered in the chronic workload by placing a greater emphasis on the most recent workloads of the 28-day cycle. The calculations were performed in Microsoft Excel. The target range of ACWR, regardless of the calculation method, was between 0.8 and 1.5 [1].

### 2.4. Statistical Analysis

All statistical analyses were performed using SPSS version 27.0 (IBM, Armonk, NY, USA). An α level of <0.05 was used to determine significance. To determine if there was a difference between the weekly totals in kinetic energy output, the RA-ACWR and the EWMAACWR, a repeated-measures analysis of variance (RM-ANOVA) was performed for both the HSVB and CVB datasets. Partial eta-squared effect sizes (ESs) were calculated to determine the magnitude of the effect. The ES values were interpreted as small (0.01), moderate (0.06), and large (0.14) [23]. If there was a main effect difference, then univariate analyses were used to evaluate specific differences within the main effect. Upon significance for a given variable, subsequently paired t-tests were used to assess weekly differences for adjacent weeks (e.g., week 3 was compared only to weeks 2 and 4) to evaluate for workload spikes and dips that would affect the fitness of the athletes. A repeated-measures correlation tests were used to evaluate the agreement between the RA-ACWR and EWMA-ACWR for the HSVB and CVB datasets [24].

## 3. Results

### 3.1. HSVB Data

The weekly average RA-ACWR, EWMA-ACWR, and workload were calculated and are shown in Figure 1. The RM-ANOVA showed a main effect difference in the weekly averages of kinetic energy, RA-ACWR, and EWMA-ACWR with a large ES—*Lambda* (39, 225) = 3.7, *p* < 0.001, ES = 0.552. The univariate analysis revealed a significant weekly difference and a large ES of the EWMA-ACWR (*p* = 0.001, ES = 0.40) and the RA-ACWR (*p* = 0.008, ES = 0.395). There was no significant difference in weekly averages of kinetic energy output (*p* = 0.141, ES = 0.288).

Paired *t*-tests showed a difference in the EWMA-ACWR between week 1 and week 2 (*t*(10) = −5.177, *p* < 0.001), week 2 and week 3 (*t*(10) = −6.376, *p* < 0.001) and week 7 and week 8 (*t*(10) = −3.004, *p* = 0.015). Paired *t*-test identified a significant difference in RA-ACWR between week 1 and week 2 (*t*(10) = −10.063, *p* < 0.001), week 2 and week 3 (*t*(10) = 6.827, *p* < 0.001). Figure 1 highlights the peak in workload in the second week of the season, followed by minor oscillations throughout the remainder of the season. The workload peak occurred in the final week of pre-season training. EWMA- and RA-ACWRs fell below the 0.8 optimal range in weeks 3 and 4 in reaction to the peak noted in week 2 and the start of the tournament phase in week 3. EWMA then increased into the optimal range by week 5, but RA did not increase until week 6.

Figure 2 shows the correlation between weekly RA- and EWMA-ACWR values for the HSVB data. The repeated-measures correlation analysis showed a strong correlation between the two metrics—r = 0.756, *p* < 0.001. Overall, the EWMA values tended to be lower than the RA values in calculating ACWR.

### 3.2. CVB Data

The weekly average RA-ACWR, EWMA-ACWR, and workload were calculated and are shown in Figure 3. The RM-ANOVA showed a main effect difference in the weekly averages of kinetic energy output, RA-ACWR, and EWMA-ACWR with a large ES—*Lambda* (42.0, 244.017) = 7.273, *p* < 0.001, ES = 0.553. The univariate analysis revealed a significant weekly difference and large ES of EWMA-ACWR (*p* < 0.001, ES = 0.642), RA-ACWR (*p <* 0.001, ES = 0.448), and kinetic energy (*p* < 0.001, ES = 0.468). All ESs were interpreted as large. The paired t-tests showed differences in the average kinetic energy output between week 3 and week 4 (*t*(7) = 3.660, *p* = 0.008), week 7 and week 8 (*t*(8) = −3.423, *p* = 0.009), week 12 and week 13 (*t*(8) = 2.845, *p* = 0.022), and week 13 and week 14 (*t*(8) = −2.308, *p* = 0.05). The EWMA-paired samples *t*-test revealed a significant difference between week 3 and week 4 (*t*(7) = 3.614, *p* = 0.009), week 7 and week 8 (*t*(8) = −3.403, *p* = 0.009), week 12 and week 13 (*t*(8) = 3.392, *p* = 0.009), week 14 and week 15 (*t*(8) = −2.343, *p* = 0.047), and week 18 and 19 (t(8) = 4.957, *p* = 0.001). The paired-sample *t*-tests demonstrated a significant difference in the RA-ACWR between weeks 3 and 4 (*t*(7) = 3.175, *p* = 0.016), week 13 was different from 12 (*t*(8) = 3.632, *p* = 0.007) and 14 (*t*(8) = −2.628, *p* = 0.03), and week 18 was different from week 17 (t(8) = −5.632, *p* < 0.001) and 19 (*t*(8) = 7.754, *p* < 0.001). Figure 3 shows that weeks 4 and 13 saw a significant decrease in workload, and in both cases, RA- and EWMA-ACWRs dropped below the 0.8 threshold to maintain fitness. This occurred a third time in week 19. Week 4 fell near the holidays, so there were likely fewer practices that occurred, and weeks 13 and 19 were during the qualifier section of the season.

Figure 4 shows the correlation between RA- and EWMA-ACWR calculations for the CVB data. The repeated-measures correlation analysis indicated a moderate correlation between the two metrics—r = 0.646, *p* < 0.001. Like the HSVB data, EWMA values tended to be lower than RA values.

## 4. Discussion

This study investigated the fluctuations in ACWR and kinetic energy load in high school–aged volleyball players, examining both the RA and EWMA methods of calculation during the high school volleyball season and the club season. HSVB demonstrated a baseline level of ACWR that was kept in the optimal range for the season, while CVB displayed many peaks and valleys that were outside of the optimal range of 0.8–1.5 [1]. The results support the use of ACWR in managing load during both volleyball seasons and that the two calculations for ACWR show higher agreement with a more consistent competition and training schedule.

### 4.1. ACWR in High School Volleyball Season

It is important to maintain fitness while preventing overtraining because the high school season is four months long, with training or games six days a week. While there was no significant difference between weekly kinetic energy averages, there were weekly differences for both RA and EWMA with large effect sizes. This suggests the need to evaluate training load in absolute terms—kinetic energy—and in reference to the chronic workload accrued through ACWR to provide a holistic view of the athlete’s load. Both methods of ACWR calculations demonstrated that coaches maintained an effective training schedule by maintaining the ACWR values within the optimal range (0.8–1.5), and the different ACWR methods were strongly correlated. The weekly ACWR values were similar to those from research on elite female volleyball players [10]. The EWMA method detected a significant spike in ACWR training between the seventh and eighth week of the season (Figure 1), while the RA method did not. The EWMA method is known for being more sensitive to spikes in training, leading to better load monitoring and possibly injury prevention [4]. Spikes in training are important to monitor to prevent injury from overtraining [1]. In addition, the EWMA method tended to result in lower values, suggesting that the RA method gives too much weight to workloads that no longer have the same effect as more recent workloads. This negatively limits the calculation’s ability to account for fitness and fatigue over time [4,22,25]. This also suggests that using the RA method can lead to erroneously lighter workload training sessions, which can result in loss of fitness and cause spikes during competition [4]. For these reasons, the EWMA method appears to be a more appropriate ACWR for a season with consistent training and competition.

### 4.2. ACWR in Club Volleyball Season

Club volleyball is typically a seven-month season. This dataset only investigated the first five months of the season because of increased training and competition inconsistency in the final two months of the season. The club season is distinguished by the two practices per week and tournament-style matches, where athletes typically compete Friday–Sunday with three to four matches per day. Additionally, these athletes might also be participating in other high school sports or an off-season volleyball program at their high school, which would also contribute to the athletes’ overall physiological load. With this type of training, it is very important to monitor workload because this is the longest season and overuse injuries can result from rapid fluctuations in workload.

The club volleyball season showed weekly differences in kinetic energy, RA, and EWMA. All three metrics showed a decline in workload between weeks 3 and 4 and weeks 12 and 13. Week 4 was very low comparatively because it aligned with a holiday break in training. A spike in training was detected based on kinetic energy and the EWMA method between weeks 7 and 8. Kinetic energy and the RA method indicated a significant spike in workload between weeks 13 and 14, while the EWMA method showed a spike between weeks 14 and 15. As EWMA is more sensitive to determining spikes, this was an expected result [4]. Both methods exhibited ACWRs that fell below the optimal range during week 4 and achieved very similar values for week 15, nearing the high end of the optimal value, indicating that the athletes could have been at risk of fatigue and injury [1]. This was most likely due to the inconsistent schedule of the club volleyball season, which is different from an elite volleyball schedule that produces a more consistent training regimen [10,26]. The EWMA method did not identify week 13 as a potential area of losing fitness, while the RA method did. This was most likely explained by the fact that the RA method weighs each weekly workload equally, causing the estimates to be linear and unable to accurately account for the variations that occur during the club schedule [4,22,25]. Interestingly, the RA method identified week 9 as possibly going over the optimal range, while the EWMA method identified the increase in week 10. This is significant because, looking at the kinetic energy, we can see that there was an increase in workload during week 9, which correlates with the RA method. However, when the kinetic energy output decreased during week 10, the EWMA picked up on the near-injury possibility level because of an increased weight on the week 9 workload. This exemplifies the EWMA’s ability to apply the time decay, allowing it to better understand the workload [4]. The moderate correlation between the two ACWR measures suggests that the methods are useful in monitoring athlete workload but provides few answers as to which method is more effective. However, when considering the formulas for each method and the inconsistent club season schedule, the importance of a time constant increases because there are many off-days during the week. Because the EWMA is more sensitive when detecting spikes and the schedule is inconsistent, it is important to be able to account for decaying fitness and fatigue when monitoring players’ workload. These results indicate the EWMA method may be a better monitoring method, but more research on seasons with inconsistent schedules needs to be conducted.

### 4.3. Application of ACWR in Girls’ Volleyball

Currently, there is no similar study that uses ACWR in female youth volleyball players, and there is no standard for load monitoring in this population. These data add to the little research available for workload assessment in volleyball across a season [11,13]. An important finding of this study is that ACWR can and should be used to monitor female athletes. The consistency of training in a high school volleyball season lends itself to the EWMA calculation, while there is not a definitive answer to which method is more appropriate for the inconsistent season of club volleyball. Future research should focus on this season type because there are many youth athletes that participate in club sports. Ratings of perceived exertion are a valid, simple, and inexpensive method of evaluating workload that can be employed with an inconsistent season [27]. While there is reliability to these methods, perception is not consistent within or between subjects, teams, or sports, and it is hard to determine validity [28,29].

Using the EWMA method for girls’ volleyball during the high school season is consistent with the current ACWR literature and is most likely due to the similar training schedule [4,5,30]. However, the club season is less predictable and consistent. While it is appropriate and consistent with the current literature that ACWR is an effective method of monitoring workload [31,32,33], there is a staggering lack of evidence for girls’ sports in general and particularly girls’ club sports.

### 4.4. Limitations

Limitations in this study were related to data availability. The final two months of the CVB season were not included in this study because of inconsistent data. This decision was made because it was the slowest training period of the season, and the data that were included in analyses comprised the densest training and playing portion of the season. In addition, compliance with the VERT monitoring system precipitated incomplete data capturing. This was controlled by removing athletes that had less than 50% participation. Another limitation of this study was that different athletes were followed through the two seasons. Ideally, the same athletes would have been followed during the training year, but this was a significant logistical challenge. Lastly, there is virtually no comparable study of girls’ volleyball players using ACWR as a workload monitoring tool. Future investigations of ACWR should include this as an important population of interest.

## 5. Conclusions

This is a novel research project that is a first attempt at characterizing the workload of girls’ youth volleyball using ACWR. These results demonstrate that, for high school volleyball, ACWR is a useful method of load monitoring and EWMA may be a more appropriate calculation method. The club volleyball season also demonstrated that ACWR is an insightful method for evaluating workload, but future research should focus on further elucidating which ACWR method is more appropriate. The inconsistent training schedule may be part of the reason there is not a definitive answer to which ACWR method seems more sensitive and a more useful monitoring tool.

## Figures and Tables

**Figure 1 sports-11-00051-f001:**
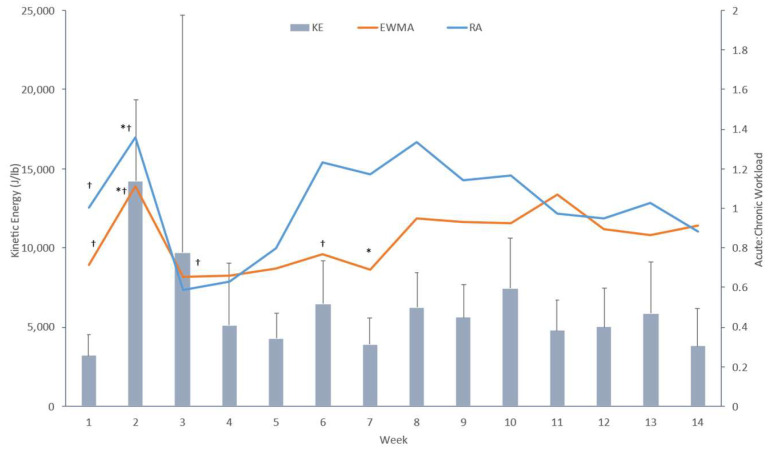
Means and standard deviations of the weekly average kinetic energy (KE, gray bar), weekly exponentially weighted moving average ACWR (EWMA, square, blue line), and weekly rolling average ACWR (RA, circle, red line) of HSVB. Kinetic energy is associated with the primary y-axis, and both ACWR lines are associated with the secondary y-axis. Dagger (†) represents significance to the week following; the asterisk (*) represents significance to the week prior.

**Figure 2 sports-11-00051-f002:**
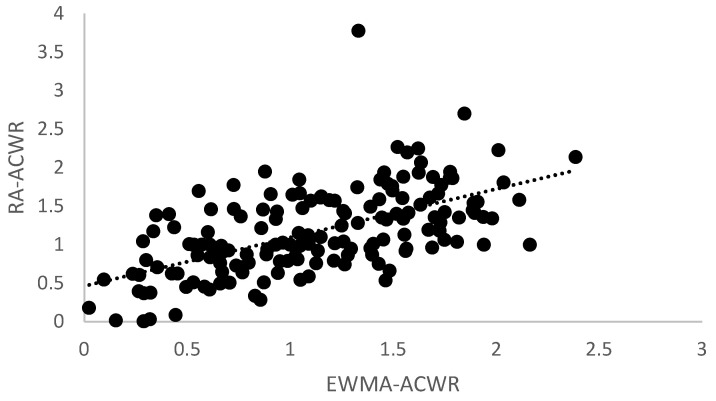
Correlation between the two ACWR metrics for the HSVB data—r = 0.756, *p* < 0.001. Rolling average (RA) vs. exponentially weighted moving average (EWMA).

**Figure 3 sports-11-00051-f003:**
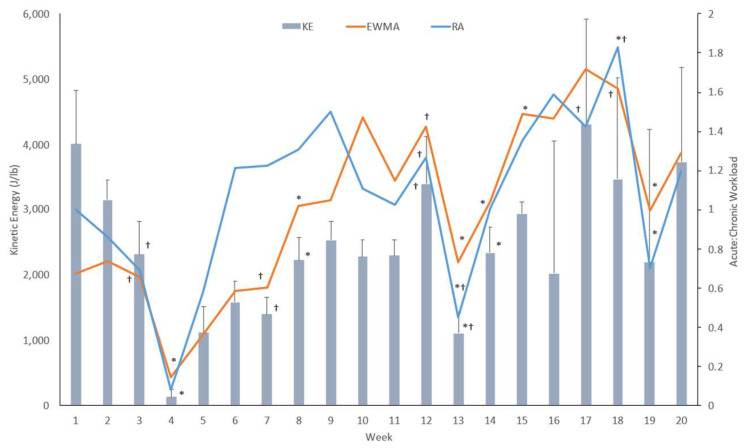
Means and standard deviations of the weekly average kinetic energy (KE, gray bar), weekly exponentially weighted moving average ACWR (EWMA, square, blue line) and weekly rolling average ACWR (RA, circle, red line) of CVB. KE is associated with the primary y-axis, and both ACWR lines are associated with the secondary y-axis. Dagger (†) represents significance to the week following, and the asterisk (*) represents significance to the week prior.

**Figure 4 sports-11-00051-f004:**
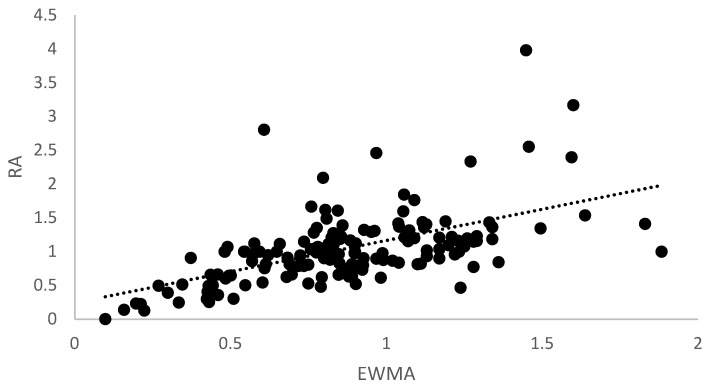
Correlation between the two ACWR metrics for the CVB data—r = 0.646, *p* < 0.001. Rolling average (RA) and exponentially weighted moving average (EWMA).

## Data Availability

The data presented in this study are openly available in Mendeley Data, V1 at doi: 10.17632/8hk8k44y7p.1. [34].

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
