# Peer review of "Comparing Two Methods of Acute: Chronic Workload Calculations in Girls’ Youth Volleyball"

_sports, 2023, doi:10.3390/sports11030051_

Round 1

Reviewer 1 Report

I do not think that this research addresses an original problem. There are many researches on this subject. The innovativeness of using only the KE parameter in the calculation is a controversial point of view.

I think it is necessary for the authors to convince the editor and the reviewers about the originality of the research. If I am convinced, I would like to review it once more.

I recommend that you review the references on the subject (EWMA and ACRWR);

https://bmcsportsscimedrehabil.biomedcentral.com/articles/10.1186/s13102-022-00568-1

https://bjsm.bmj.com/content/51/9/749.short

https://journals.sagepub.com/doi/abs/10.1177/17543371221101233

https://www.frontiersin.org/articles/10.3389/fphys.2020.00608/full

https://bjsm.bmj.com/content/51/3/209.short

Reviewer 2 Report

The authors of this manuscript aimed to compare changes in external load experienced by young female volleyball players during the highschool and club seasons. They decided to measure/estimate that load by using the notion of kinetic energy expressed in J/lb, thus in fact kinetic energy per unit of body weight. The only place in the text where they explain how this quantity was evaluated are lines 143 and 144, which read "Kinetic energy was calculated by multiplying the total time of the session and the total acceleration of the athlete". This sounds rather enigmatic and may mislead readers, because multiplying time and acceleration leads to something expressed in units of velocity (m/s), whereas J/lb behaves similarly to J/kg which is equal to (m/s)2. So I suspect there is something wrong with this definition.

Perhaps the authors actually applied the definition of kinetic energy (KE) often used in similar studies, which was proposed in reference (1) below (Charlton et al. 2017, page 243, left column, lines 7-18 counting from top). If so, I would recommend adding that reference to the reference list, and providing a better description of how the main quantity (KE) used for comparisons was evaluated.

Another important paper potentially relevant to this study is reference (2) below (Brooks et al. 2021). Please consider adding it as well.

(1) Charlton, Kenneally-Dabrowski, Sheppard, Spratford, A simple method for quantifying jump loads in volleyball athletes

DOI:10.1016/j.jsams.2016.07.007

(2) Brooks, Benson, Fox, Bruce, Quantifying jumps and external load in netball using VERT inertial measurement units

https://doi.org/10.1080/14763141.2021.2009906

Lines 39, 40: "model does not take time into account" should possibly read "model does not take into account"

Lines 111, 112: "divided into four phases", and four phases follow

but 

Lines 124, 125: "divided into three phases", and four phases follow, please check if this is correct

Lines 103 and 365: please check if the distinction between assent and consent is used consistently throughout the manuscript

Round 2

Reviewer 1 Report

As I mentioned before, there is no significant innovation in this research. I congratulate the researchers for their efforts, but I do not think they will significantly contribute to the field. The editor is free to decide just because they are in high school age. However, the fact that the population is different does not make the research unique.
